# 3DOT: Texture Transfer for 3DGS Objects from a Single Reference Image

**Xiao Cao**[1]  **Beibei Lin**[1]  **Bo Wang**[2]  **Zhiyong Huang**[1]  **Robby T. Tan**[1,3]

[1] National University of Singapore   [2] University of Mississippi   [3] ASUS Intelligent Cloud Services

{xiaocao, beibei.lin}@u.nus.edu

hawk.rsrch@gmail.com   {dcshuang,robby.tan}@nus.edu.sg

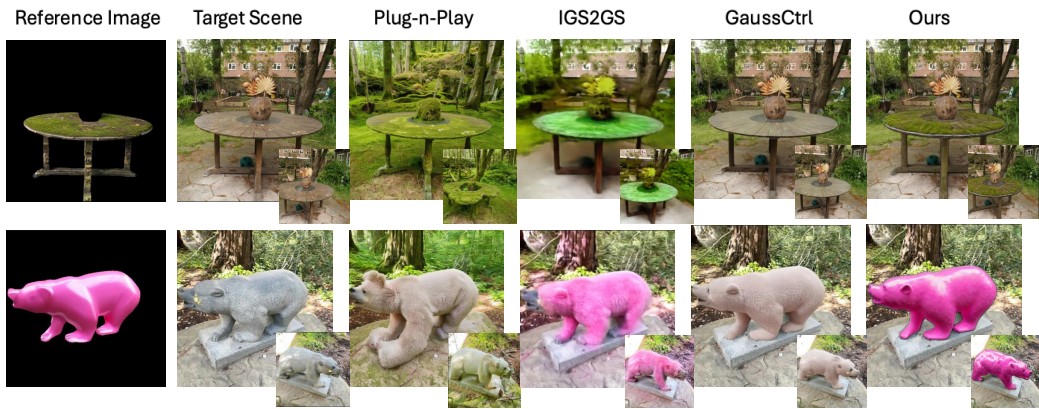

Figure 1: Comparison of 2D and 3D image-based texture editing methods. Prompts are "**moss-covered table**" and "**pink plastic bear**". 2D methods Plug-n-Play [36] suffers from view inconsistency problem; 3D text-driven editing methods IGS2GS [37] and GaussCtrl [41] struggle to preserve texture characteristics. Ours faithfully edit the texture, material appearance, and color.

## Abstract

Image-based 3D texture transfer from a single 2D reference image enables practical customization of 3D object appearances with minimal manual effort. Adapted 2D editing and text-driven 3D editing approaches can serve this purpose. However, 2D editing typically involves frame-by-frame manipulation, often resulting in inconsistencies across views, while text-driven 3D editing struggles to preserve texture characteristics from reference images. To tackle these challenges, we introduce **3DOT**, a **3D** Gaussian Splatting **O**bject **T**exture Transfer method based on a single reference image, integrating: 1) progressive generation, 2) view-consistency gradient guidance, and 3) prompt-tuned gradient guidance. To ensure view consistency, progressive generation starts by transferring texture from the reference image and gradually propagates it to adjacent views. View-consistency gradient guidance further reinforces coherence by conditioning the generation model on feature differences between consistent and inconsistent outputs. To preserve texture characteristics, prompt-tuning-based gradient guidance learns a token that describes differences between original and reference textures, guiding the transfer for faithful texture preservation across views. Overall, **3DOT** combines these strategies to achieve effective texture transfer while maintaining structural coherence across viewpoints. Extensive qualitative and quantitative evaluations confirm that our three components enable convincing and effective 2D-to-3D texture transfer. Our project page is available here: `https://massyzs.github.io/3DOT_web/`.

39th Conference on Neural Information Processing Systems (NeurIPS 2025).

# 1 Introduction

Transferring texture from a 2D image to a 3D object is a valuable yet underexplored capability in 3D editing. It enables efficient texture manipulation and benefits applications such as virtual reality, CG films, and 3D games [2, 31, 40, 23, 28, 39, 34, 5]. Despite advances in 2D texture and 3D editing techniques, transferring texture from a single 2D image to a 3D object remains challenging due to difficulties in ensuring view consistency and preserving texture characteristics, particularly for unseen views beyond the reference image.

*2D image-based editing* methods [36, 46, 25, 32, 14, 51, 24, 42] perform texture transfer by finetuning a diffusion model (e.g., DreamBooth [30], Textual Inversion [13]) and editing images rendered from a 3D object to create a finetuning dataset. The resulting 3D object often suffers from view inconsistency and identity loss due to the absence of constraints enforcing multi-view coherence and identity preservation, as shown in Figure 1. *3D editing* methods [15, 37, 9, 41, 8, 27, 6], especially text-driven ones, guide editing using prompts derived from reference images via visual language models or manual descriptions. However, these prompts are typically coarse and miss fine-grained features, resulting in identity mismatch and inconsistent appearance across views.

Motivated by these challenges, we propose **3DOT**, a novel framework for transferring texture from a single 2D reference image to a 3D object represented by 3D Gaussian Splatting [21]. *3DOT* comprises three key components: 1) a progressive generation process, 2) view-consistency gradient guidance, and 3) prompt-tuning-based gradient guidance. The first two components enforce view consistency, while the third preserves texture characteristics.

In the progressive generation process, we first obtain reference images either by directly pasting the reference image onto the 3D object or by generating candidate views using a depth-conditioned model [47] based on the unedited view's depth. The image that best matches the target attributes is then selected. To facilitate prompt tuning and sparse cross-attention, we remove backgrounds from both the unedited training images and the reference images, and project them into the latent space for k-step partial diffusion. The generation begins from the reference view and progressively propagates to neighboring views, guided by sparse cross-attention on previously edited views. This strategy maximizes overlap between adjacent reference images to enforce view consistency.

To enhance view consistency in 3D editing, we introduce view-consistency gradient guidance. The core idea is to guide the diffusion model toward view-consistent generation by minimizing texture inconsistency features in intermediate outputs. Specifically, we initialize two diffusion modules: one conditioned on reference views via cross-attention, and the other guided only by a text prompt. Since cross-attention is the only differing component, the discrepancy between their intermediate results captures view-consistency features. During each denoising step, these features are scaled and injected as gradient guidance, steering the generation toward consistent outputs across views.

Since the reference image reveals no texture for unseen views, coarse text prompts often lead to inconsistency. To overcome this, we propose prompt-tuning-based gradient guidance that captures texture differences as additional prompt tokens. Specifically, we compute the difference between reference and unedited images in the CLIP feature space [11], encoding the texture transformation direction. This signal is injected into the diffusion denoising process as gradient guidance, enabling consistent texture transfer across views. The fine-tuned prompt improves style coherence in unseen views while preserving details in the reference view.

We evaluate our method on the face-forwarding [38] and 360-degree [3] datasets. Results show effective texture transfer with fine detail preservation and strong view consistency. Our key contributions:

- **3DOT**, an image-based 3D Guassian Splatting (3DGS) texture transfer framework that enables efficient and flexible texture editing.

- A progressive generation process with view-consistency gradient guidance to address view inconsistency across novel views.

- Prompt-tuning-based gradient guidance preserves texture characteristics in seen views and enforces style consistency in unseen views.

- Extensive experiments demonstrate that **3DOT** achieves state-of-the-art visual quality and quantitative performance.

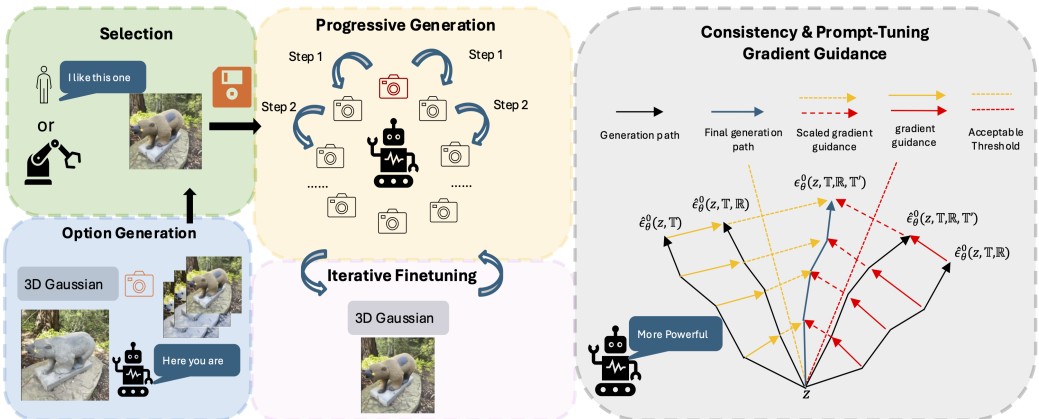

Figure 2: **3DOT.** Our framework enables texture transfer from a single image to a 3D object. The left panels illustrate the selection of the reference image using a generative approach. Then, our method employs a progressive generation process guided by view-consistency and prompt-tuning-based gradient guidance to preserve both cross-view consistency and texture identity. $\mathbb{R}$, $\mathbb{T}$, and $\mathbb{T}'$ denote the reference set, text prompt, and learned texture difference token, respectively.

## 2 Related Work

**2D Diffusion-based Editing** DragDiffusion [32] defines target edits using keypoints and replaces them with reference images. A-Tale-of-Two-Features [46] combines dense DINO [7] features and sparse diffusion features by merging reference-view semantics with target-view structures. Plug-and-Play [36] refines fine-grained details by injecting diffusion features into DINO features, while DiffEditor [25] improves 2D editing precision via differential equation-based sampling with regional gradient guidance. The most relevant work, SwapAnything [14], leverages DreamBooth [30] and AdaIN [19] to encode source images and maintain style consistency during 2D edits. Although effective for image editing, these methods operate on individual views without enforcing view consistency, highlighting the need for 3D-aware texture editing techniques.

**3D Editing** Most 3D editing methods leverage 2D diffusion models for guidance and adopt dataset-updating strategies to finetune pretrained 3D scenes. Instruct-NeRF2NeRF [15] and Instruct-GS2GS [37] use instruct-pix2pix [4] to guide updates for NeRF or Gaussian Splatting. GaussianEditor [9] introduces hierarchical representations for more stable edits under stochastic guidance. Direct Gaussian Editor (DGE) [8] addresses view consistency via epipolar cross-attention, but its initial independent generation introduces artifacts. GaussCtrl [41] injects features from unedited views to preserve consistency, but this can cause the diffusion model to retain original textures, limiting editability. StyleSplat [20] achieves texture edits without a generative model but requires altering the 3D representation, which falls outside our setting of editing a fixed 3D object using a single reference image. Methods that ignore view consistency [15, 37, 9] can be extended with image captioning, while consistency-aware approaches [8, 41] can inject latent reference features during denoising. However, such modifications offer only coarse control. High-quality, identity-preserving edits require more precise and targeted designs.

## 3 Proposed Method

Fig. 2 illustrates our **3DOT** pipeline, consisting of three key modules: 1) a progressive generation process, 2) view-consistency gradient guidance for enforcing texture coherence across different views, and 3) prompt-tuning-based gradient guidance for preserving object identity.

To obtain the reference image, we either generate depth-conditioned candidates or extract textures by directly cropping texture into object shape in a certain rendered view. In the generative approach, users select the candidate that best matches the desired attributes. In the texture-based approach, extracted textures are directly mapped onto the object surface. Following [41], both reference and unedited images are encoded into latent space to initialize the denoising process. We then apply

prompt-tuning to capture texture differences between the reference and the 3D object, guiding diffusion to preserve identity. Edited views are progressively generated, starting from the reference view. The resulting dataset is utilized to finetune 3D Gaussian model and the above procedure is iteratively conducted [15] for smooth results.

## 3.1 Progressive Generation

Existing methods [15, 37, 9, 41, 8] struggle to balance view consistency and editing flexibility. For example, GaussCtrl [41] conditions diffusion on unedited images to enforce consistency but often retains original textures, limiting editability. DGE [8] avoids reliance on unedited inputs but edits non-adjacent views, introducing inconsistencies.

To overcome these limitations, we propose a progressive generation process that removes dependency on unedited images and avoids isolated generation steps, achieving both consistency and flexibility.

We first generate reference images by conditioning a generative model on depth maps with background masking to ensure geometric alignment. To improve quality, we refine depth maps using dilated and blurred masks to address black-edge artifacts and apply the original mask to remove redundant content from the outputs.

For a selected reference view $\tau$ and target view $\mathbb{I}_i$, we construct a sparse reference set $\mathbb{R}_i = \{\mathbb{I}_\tau, \mathbb{I}_{i-1}, \mathbb{F}(\mathbb{I})_\tau\}$, excluding backgrounds. Including $\mathbb{I}_{i-1}$ maintains local consistency via minimal angular changes. As edits propagate to distant views, errors accumulate. For symmetric case, we can include $\mathbb{F}(\mathbb{I})_\tau$, a horizontally flipped variant of the reference, to preserve alignment with fewer conditioning views.

The generative model is conditioned on $\mathbb{R}$ using weighted fused cross-attention:

$$\text{WeightedAttn}_e = \lambda \text{Attn}_{e,e} + (1 - \lambda) \sum_{i \in \mathbb{R}} w_i \text{Attn}_{e,i}, \tag{1}$$

where $e$ denotes the image that is currently editing, $\text{Attn}_{i,j}$ denotes the attention score between images $i$ and $j$, and $\lambda$ balances self- and cross-attention.

Partial denoising (Eq. 3) begins with the reference view and progressively extends to adjacent views. These edited, view-consistent images are then used to finetune the 3D Gaussian model, and the process is repeated iteratively.

## 3.2 View-Consistency Gradient Guidance

Existing generative methods [47, 45, 30, 13] rely on many reference views to maintain consistency. In contrast, our progressive generation begins with a single reference and uses only a few views. To enhance cross-attention effectiveness under this constraint, we propose a consistency-aware gradient guidance mechanism inspired by classifier-free guidance [16], modifying the noise estimate [16] to amplify cross-view signals without additional training.

Given a target view $\mathbb{I}_i$ and reference set $\mathbb{R}_i = \{\mathbb{I}_\tau, \mathbb{I}_{i-1}, \mathbb{F}(\mathbb{I})_\tau\}$ (as in Sec. 3.1), we define the denoising prediction as:

$$\begin{aligned}
\epsilon_\theta^t(z_\lambda, \mathbb{T}, \mathbb{R}) = \; &\epsilon_{\hat{\theta}}^t(z_\lambda) \\
&+ w_\mathbb{T}\big(\epsilon_\theta^t(z_\lambda, \mathbb{T}, \mathbb{R}) - \epsilon_\theta^t(z_\lambda, \mathbb{R})\big) \\
&+ w_\mathbb{R}\big(\epsilon_\theta^t(z_\lambda, \mathbb{T}, \mathbb{R}) - \epsilon_{\hat{\theta}}^t(z_\lambda, \mathbb{T})\big),
\end{aligned} \tag{2}$$

where $\mathbb{T}$ is the text prompt, $\theta$ and $\hat{\theta}$ refer to diffusion with and without fused cross-attention, and $w_\mathbb{T}, w_\mathbb{R}$ are scaling factors.

We perform partial denoising as:

$$\begin{aligned}
z^{(t-1|\kappa)} = \; &\sqrt{\alpha_{t-1|\kappa}} \frac{z^{t|\kappa} - \sqrt{1 - \alpha_{t|\kappa}} \cdot \epsilon_\theta^{t|\kappa}(z, \mathbb{T}, \mathbb{R})}{\sqrt{\alpha_{t|\kappa}}} \\
&+ \sqrt{1 - \alpha_{t-1|\kappa}} \epsilon_\theta^{t|\kappa}(z, \mathbb{T}, \mathbb{R}),
\end{aligned} \tag{3}$$

| Unedited Scene | IN2N [15] | IGS2GS [37] | GaussCtrl [41] | DGE [8] | Ours | Reference Image |

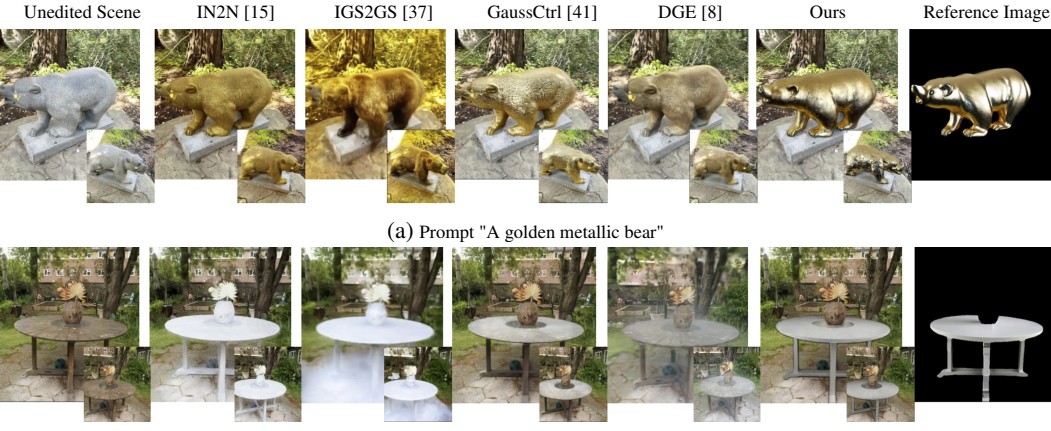

(a) Prompt "A golden metallic bear"

(b) Prompt "A white table"

Figure 3: Qualitative comparison on 360-degree scenes (material and color edits): Our 3DOT method faithfully edits 3D objects' texture based on reference images.

where $t \in [0, \kappa]$, $\kappa_\tau < \kappa_{i \neq \tau}$, and $\alpha$ is the DDIM scheduler coefficient. The latent input $z^t$ is initialized via:

$$z^{t+1} = \sqrt{\alpha_{t+1}} \frac{z^t - \sqrt{1 - \alpha_t} \cdot \epsilon^t}{\sqrt{\alpha_t}} + \sqrt{1 - \alpha_{t+1}} \epsilon^t. \tag{4}$$

In Eq. 2, the second term reflects differences between guidance with and without the unconditional prompt [16], improving adherence to textual instructions. The third term captures variations induced by reference conditioning, and amplifying it strengthens view consistency. This gradient-based mechanism guides generation toward coherent multi-view results, enhancing consistency without additional training overhead.

## 3.3 Prompt-Tuning-Based Gradient Guidance

Text prompts provide only coarse control during diffusion, often leading to identity loss and inconsistent texture fidelity. For example, the phrase "stone bear" may yield highly variable textures across different generations. These coarse text descriptions result in view-inconsistent generations and further cause 3D inconsistency.

Among fine-tuning methods [30, 45, 17, 13], textual inversion [13] learns a custom token to represent object-specific textures but requires multiple images to achieve reasonable quality (see supplementary for single-image results).

To address this, we introduce prompt-tuning-based gradient guidance, which reduces the need for multiple images while encoding texture differences more effectively. The key idea is to learn a new token that captures the texture discrepancy between the unedited 3D object and the reference image, and to use this token to guide denoising toward the desired style.

Given the reference image $\mathbb{I}_\tau$ and its corresponding unedited rendering $\hat{\mathbb{I}}_\tau$, we compute the texture difference in CLIP feature space:

$$\Delta_{\hat{\mathbb{I}}_\tau \to \mathbb{I}_\tau} = \text{CLIP}(\hat{\mathbb{I}}_\tau) - \text{CLIP}(\mathbb{I}_\tau). \tag{5}$$

We initialize the text token $\hat{\mathbb{T}}$ using a base prompt (e.g., from a VLM), and optimize it by aligning with the texture difference via:

$$L_{\text{clip}} = \text{cosine}(\Delta_{\hat{\mathbb{I}}_\tau \to \mathbb{I}_\tau}, \hat{\mathbb{T}}). \tag{6}$$

To reduce misalignment between image and text representations in CLIP space, we apply further prompt tuning in the diffusion feature space, following [26]:

$$L_{\text{diff}} = \epsilon_\theta(z_\lambda, \mathbb{T}', \mathbb{R}) - \epsilon'_\theta(z_\lambda, \mathbb{T}', \mathbb{R}). \tag{7}$$

| Metrics | IN2N | IGS2GS | GaussCtrl | DGE | Ours |
|---|---|---|---|---|---|
| CLIP Score ↑ | 0.8917 | 0.8908 | 0.8638 | 0.8572 | **0.9333** |
| Lpips(Alex) ↓ | 0.1708 | 0.1683 | 0.1692 | 0.1713 | **0.1166** |
| Lpips(VGG) ↓ | 0.1676 | 0.1594 | 0.1591 | 0.1603 | **0.1247** |
| Vision-GPT ↑ | 45.5 | 52 | 48 | 54 | **76** |
| User study ↑ | 2.0375 | 2.4375 | 2.3750 | 2.0000 | **4.5750** |

Table 1: Quantitative results evaluated by CLIP score, VGG-based and Alex-based LPIPS scores, Vision-GPT and user studies given reference image with rendered edited objects. **Bold** text refers to the best performance and underlined text refers to the second best performance. Detailed results can be found in Supplementary Material Section 2.

The fine-tuned token $\mathbb{T}'$ acts as a style-aware prompt enriched by texture differences. While not meaningful in textual form, it encodes critical style information for guiding generation. During inference, we extract and amplify this information at $t$-step via the difference:

$$\epsilon_\theta^t(z_\lambda, \mathbb{T}', \mathbb{R}) - \epsilon_{\hat{\theta}}^t(z_\lambda, \mathbb{T}, \mathbb{R}),$$

and integrate it into the denoising process. The final prediction becomes:

$$\begin{aligned}
\epsilon_\theta^t(z_\lambda, \mathbb{T}, \mathbb{R}, \mathbb{T}') = {} & \epsilon_{\hat{\theta}}^t(z_\lambda) \\
& + w_{\mathbb{T}}\big(\epsilon_\theta^t(z_\lambda, \mathbb{T}, \mathbb{R}) - \epsilon_\theta^t(z_\lambda, \mathbb{R})\big) \\
& + w_{\mathbb{R}}\big(\epsilon_\theta^t(z_\lambda, \mathbb{T}, \mathbb{R}) - \epsilon_{\hat{\theta}}^t(z_\lambda, \mathbb{T})\big) \\
& + w_{\mathbb{T}'}\big(\epsilon_\theta^t(z_\lambda, \mathbb{T}', \mathbb{R}) - \epsilon_{\hat{\theta}}^t(z_\lambda, \mathbb{T}, \mathbb{R})\big).
\end{aligned} \tag{8}$$

This term strengthens style consistency across views while preserving fine texture details aligned with the reference.

## 4 Experiments

We compare our method with state-of-the-art text-driven editing approaches, including GaussCtrl [41], DGE [8], IGS2GS [37], and IN2N [15]. Since these methods rely on text inputs, we use captioned descriptions as editing prompts to enable image-based 3D texture editing functionality. For quantitative evaluation, we employ AlexNet-based [22] and VGG-based [33] LPIPS scores [48], CLIP score [29], and Vision-GPT score [1], supplemented by user studies. Comparisons are conducted across multiple scenes from different datasets to ensure a comprehensive assessment following [41].

### 4.1 Evaluation

**Quantitative** For each edit, we compute AlexNet-based and VGG-based LPIPS scores, CLIP score, Vision-GPT score, and conduct user studies, as summarized in Table 1. Detailed per-scene scores are provided in the supplementary material. LPIPS and CLIP scores serve as perceptual evaluation metrics, measuring feature similarity between rendered edited objects and reference images. LPIPS ranges from 0 to 1, with lower values indicating better perceptual quality, while higher CLIP scores are preferred. Vision-GPT assesses the faithfulness of edited textures from the reasoning perspective, scoring from 0 to 100, where higher values indicate better alignment. For user studies, participants are informed of the edited object and required to rate the 3D result on a scale of 1 to 5, with higher scores reflecting better quality. Quantitative results show that our method achieves the highest performance across all metrics.

**Qualitative** We present qualitative results of 360-degree dataset in Figs.1, 3 and 4. Figs.3 and 3 includes reference images with texture color or material variations, while Fig.4 features those with complex textures and significant semantic changes. Fig.5 shows the results of "face-forward" case. Our method enables more precise 3D object editing without unintended texture leakage between objects. In the 360-degree color and material editing scenario (e.g., bear and table), IN2N [15] and IGS2GS [37] suffer from incorrect color saturation and inaccurate material representation. In the bear scenarios (Fig.1, 3a), their results are undersaturated, whereas in the table scenarios (Fig.3b, 1),

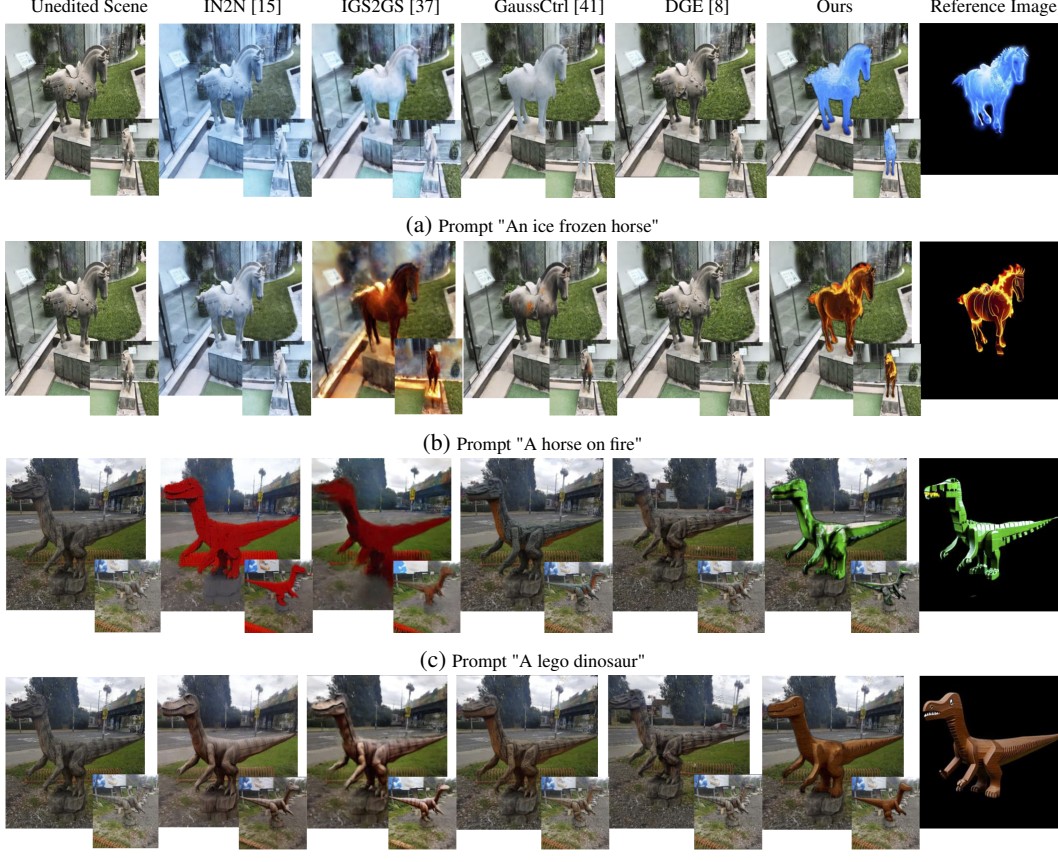

Unedited Scene    IN2N [15]    IGS2GS [37]    GaussCtrl [41]    DGE [8]    Ours    Reference Image

(a) Prompt "An ice frozen horse"

(b) Prompt "A horse on fire"

(c) Prompt "A lego dinosaur"

(d) Prompt "A wooden dinosaur"

Figure 4: Qualitative comparison on 360-degree scenes (complicated texture edits): Our 3DOT method successfully edits 3D objects' texture to complicated reference textures.

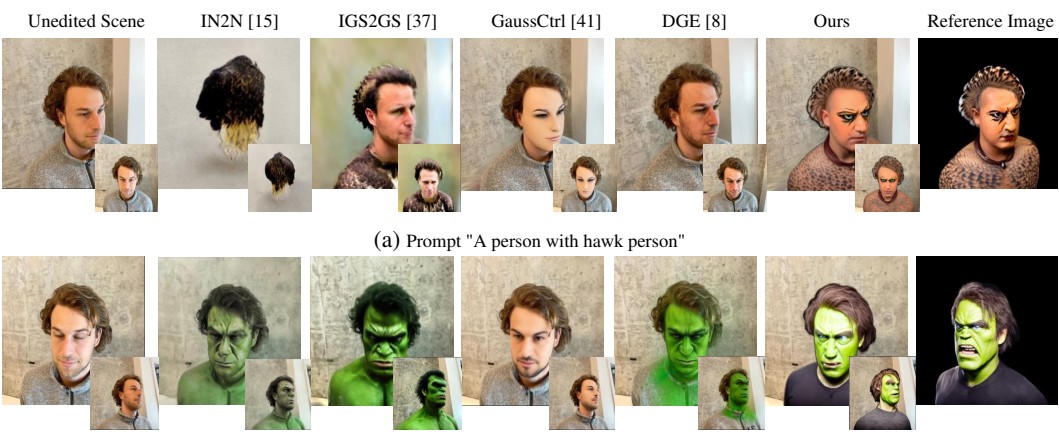

Unedited Scene    IN2N [15]    IGS2GS [37]    GaussCtrl [41]    DGE [8]    Ours    Reference Image

(a) Prompt "A person with hawk person"

(b) Prompt "A person with Hulk face"

Figure 5: Qualitative comparison on face-forwarding scenes: Our 3DOT method faithfully edits 3D objects' texture to reference textures and generates the most plausible texture edits for unseen views.

they are over-saturated. None of the baseline methods accurately reproduces the intended material attributes (i.e., plastic, moss in Fig. 1 and metallic in Fig. 3a). GaussCtrl [41] excessively preserves the original 3D object's appearance, resulting in minimal modifications due to its unedited reference set. Our method effectively edits textures while achieving realistic material appearances, such as

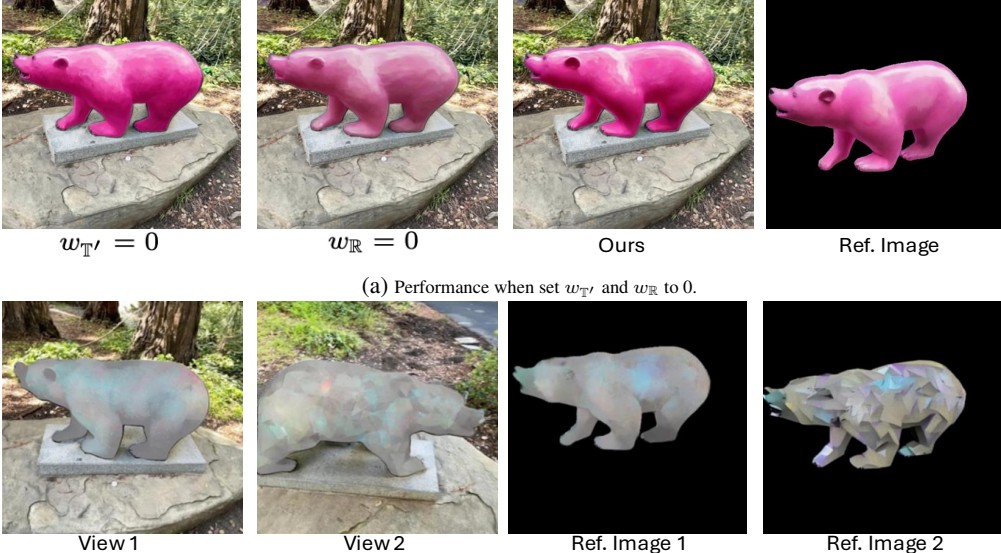

(a) Performance when set $w_{\mathbb{T}'}$ and $w_{\mathbb{R}}$ to 0.

(b) Using Ref.1 for prompt-tuning guidance and Ref.2 as reference image

Figure 6: Ablation studies on proposed two gradient guidances.

specular highlights on the bear and the lush, velvety moss on the table. In *moss-covered table* scenario (Fig.1), all 3D baseline methods only attempt to edit texture while ours can also modify geometry to better match the "moss material".

In the large semantic change editing scenario (Fig.4), baseline methods struggle with significant transformations. DGE [8] often fails, as its edits remain nearly unchanged. Its initial independent editing stage leads to inconsistent results and further causes the epipolar attention mechanism to break down in highly dissimilar views, resulting in minimal overall changes. Our method achieves precise 3D object editing with a texture style that closely matches the reference image, enabled by our proposed prompt-tuning, consistency guidance, and progressive process. Prompt-tuning preserves intricate texture details, while consistency guidance and progressive generation mitigate blurriness from view inconsistency.

In the face-forward case (Fig.5), our method preserves fine details, such as the black lower eyelid and feather-like cloth in the hawk scenario (Fig.5a). IN2N [15] and IGS2GS [37] generate erroneous results due to their independent diffusion process and full diffusion steps. The independent generation process leads to inconsistent images, while full diffusion steps cause excessive texture changes and identity loss. Finetuning NeRF with these inconsistent and identity-lost images can result in network collapse. For GaussCtrl [41] and DGE [8], particularly in the hawk case, large texture differences break their view-consistency mechanisms, resulting in outputs that retain the original object's appearance instead of the intended modifications.

## 4.2 Editing Speed

We compare the editing time of our method with two baselines: the Gaussian Splatting model (GaussCtrl [41]) and a representative NeRF-based model (IN2N [15]). GaussCtrl requires 15min 47s, while IN2N takes 20h 51min 20s. Our approach preserves the efficiency advantage of Gaussian Splatting, with an editing time of 23min 33s, introducing only a modest additional overhead from the incorporation of view-consistency and prompt-tuning-based guidance. In particular, the additional time required by our guidance components is approximately 8min in total. Since our full pipeline typically involves two iterations of image editing, the added cost per iteration is about 4min. We consider this overhead efficient given the improvements in texture fidelity and view consistency achieved by our method.

| Ablation | LPIPS (Alex)↓ | LPIPS (VGGT)↓ | CLIP Score↑ |
|---|---|---|---|
| W/O prompt-tuning (Sec.3.3) | 0.0355 | 0.0679 | 0.9203 |
| W/O consistency (Sec.3.2) | 0.0558 | 0.0844 | 0.9340 |
| W/O Prog. Gen. | 0.1330 | 0.1290 | 0.9160 |
| Ours | **0.0351** | **0.0620** | **0.9445** |

Table 2: Ablation studies: Performance evaluation when removing (1) prompt-tuning guidance, (2) view-consistency guidance, and (3) progressive generation mechanism.

## 4.3 Ablation Studies

We evaluate the effectiveness of prompt-tuning-based gradient guidance and view-consistency gradient guidance from both qualitative and quantitative perspectives. We further illustrate the effectiveness of prompt-tuning-based guidance by using two distinct images: one serving as the reference image during image generation, and the other as the reference for prompt tuning.

**Prompt-tuning based Gradient Guidance**     We first evaluate the effect of removing prompt-tuning-based guidance by setting the guidance scale $w_{\mathbb{T}'} = 0$, as shown in Table 2 and Figure 6a. Without this guidance, the rendered images exhibit blurry surface highlights, and the distribution direction does not align with the reference image.

Additionally, we demonstrate the effectiveness of prompt-tuning guidance by using a prompt token trained on reference image Ref-2 to edit the 3D object and setting Ref-1 as the target reference, as illustrated in Figure 6b. Ref-2 depicts a bear characterized by sharp metallic edges, whereas Ref-1 shows a bear with a rusted metallic texture. In Figure 6b, View-1 aligns closely with Ref-1, producing a rendering that closely matches the rusted metal appearance, which demonstrates the effectiveness of the utilized fused cross-attention. Conversely, View-2, which represents an unseen viewpoint without explicit texture guidance, utilizes the learned token to guide the rendering towards the "colorful metallic bear" appearance consistent with Ref-2. This demonstrates the effectiveness of the proposed prompt-tuning-based guidance when editing parts of the 3D object not visible in the reference image.

**View-consistency Gradient Guidance**     We evaluate the effectiveness of view-consistency guidance by setting $w_{\mathbb{R}} = 0$, as reported in Table 2 and illustrated in Figure 6a. When this guidance is disabled, the performance significantly degrades, resulting in edited images exhibiting notable undersaturation. This undersaturation primarily arises due to inconsistencies within overlapping regions in the intermediate outputs. These findings underscore the crucial role of view-consistency gradient guidance in maintaining editing quality and color fidelity.

**Progressive Generation**     To evaluate the effectiveness of the progressive generation, we disable progressive view propagation and perform editing using only the initial reference image. The degraded performance (as shown in Table 2) highlights the importance of propagating texture across neighboring views to enforce view consistency and mitigate artifacts from single-view editing.

## 4.4 Discussion

**Differences from Multi-View Diffusion**     While existing multi-view diffusion methods are designed to generate multi-view consistent 3D objects, the task setting, constraints, and diffusion components in our work are fundamentally different. Specifically, **3DOT** takes as input a single 2D reference image and a fixed 3D Gaussian Splatting (3DGS) representation, and transfers high-fidelity texture onto this existing geometry. In contrast, multi-view diffusion methods are typically designed to synthesize novel views or reconstruct 3D scenes from a few texture-consistent input images, without anchoring to an explicit 3D representation. These methods rely on implicit geometry learned from priors, which makes them unsuitable for editing tasks that require consistency with a given 3D structure. Moreover, multi-view diffusion approaches assume that texture appearance is consistent across views, an assumption that does not hold in texture transfer scenarios where the reference image and the 3D representation exhibit different textures. Directly applying them in this setting leads to collapsed reconstructions, highlighting the need for a specialized framework such as ours.

**Different Geometric Reference Image**    Geometry mismatches between the 2D reference image and the 3D object can occur when the reference is generated via a depth-conditioned diffusion model with a small conditioning scale factor. We consider two common scenarios:

- **Slightly Different Geometry:** Minor deviations (e.g., slight pose or scale differences, such as a standing bear with legs in a different position) can be effectively handled by **3DOT**. (1) During progressive generation, cross-attention extracts texture features while being constrained by the depth map and partially reversed latent features, preserving appearance cues from the original image. (2) During 3D fine-tuning, overlapping regions from adjacent edited views iteratively correct geometric discrepancies and reinforce consistent texture transfer.

- **Significantly Different Geometry:** When the reference depicts a substantially different shape (e.g., a running bear instead of a standing one), transfer quality may degrade. Such cases usually result from failures in depth-conditioned option generation or from unsuitable user-provided references. In practice, these poor references can be easily identified during the interactive selection stage, and regenerated using our partial denoising strategy with a larger depth control factor.

In summary, **3DOT** is robust to minor geometric mismatches, and mitigates larger discrepancies through reference regeneration and iterative correction during 3D fine-tuning.

**Limitation**    Dark border around edited regions occurs in some cases. We attribute this artifact to language-based object segmentation methods (e.g., LangSAM) to generate masks for isolating objects in the reference views. These segmentation methods often include a narrow band of background pixels near object boundaries due to imperfect boundary localization. As a result, during the diffusion-based editing stage, this narrow band is misinterpreted as valid texture content, leading to the appearance of dark borders in the final renderings even after applying the soften mask. This may be addressed by utilizing a more advanced language-based segmentation method [49, 43, 35], depth as additional information [44, 50, 10] with a semantic 3D representation [18, 12]. The quality of unedited 3D Gaussians also impacts editing performance. Undertrained Gaussian spheres (e.g., floating Gaussians in empty space) degrade rendered images, disrupting the mask generation process. Incorrect segmentation can result in edits with significantly altered geometry, ultimately causing 3D Gaussian collapse.

## 5  Conclusion

We introduced **3DOT**, a framework for image-based 3DGS texture transfer from a single reference image, an underexplored capability in the 3D editing domain. To enable high-quality and view-consistent texture transfer, we proposed three key components: (1) progressive generation, (2) view-consistency gradient guidance, and (3) prompt-tuning-based gradient guidance. These components effectively address challenges of view-consistency and texture characteristic preservation during transfering process. We evaluated **3DOT** across various scenes involving color, material, and large semantic changes. **3DOT** consistently outperforms existing baselines, both visually and quantitatively.

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
