# OpenReview forum: "3DOT: Texture Transfer for 3DGS Objects from a Single Reference Image"
_NeurIPS.cc/2025/Conference — NeurIPS 2025 poster_

### Official Review · Reviewer_Lzfr · 2025-06-02

**Clarity:** 4
**Significance:** 3
**Originality:** 3
**Rating:** 6
**Confidence:** 5

**Summary:**

This paper introduces 3D-OTT, an innovative approach for 3D object texture transfer using a single reference image. The method consists of a progressive generation process combined with view-consistency gradient guidance and prompt-tuning gradient guidance to ensure texture view-consistency and preservation of object characteristics.
The progressive generation method proposed is both straightforward and highly effective, enabling the generation of high-quality reference images. The authors introduce prompt-tuning gradient guidance by leveraging texture differences in the CLIP space, creating a token for fine-tuning within a diffusion framework to bridge the gap caused by mismatches between CLIP image and text features. Their approach demonstrates superior characteristic preservation compared to established techniques such as Dreambooth or Textual Inversion, as clearly indicated in the supplementary materials. Additionally, the view-consistency gradient guidance employs intermediate result differences as gradient guidance, effectively tackling view-consistency challenges in a training-free manner.

**Questions:**

1. Please elaborate on the performance differences between using tokens directly obtained from the CLIP space versus tokens that are randomly initialized and subsequently fine-tuned within the diffusion framework?
2. What is the inference speed of the proposed method in comparison to existing baseline approaches?

**Ethical Concerns:**

["NO or VERY MINOR ethics concerns only"]

**Final Justification:**

After reading the authors’ rebuttal, most of my concerns have been addressed. In addition, I believe the ablation studies on Progressive Generation and Flipped Reference Image effectively respond to the concerns raised by Reviewer ZPz8 and Reviewer Eep3. Taking all aspects into account, I find this work sufficiently solid. Based on the authors’ response and the comments from other reviewers, I have decided to raise my score and recommend acceptance.

**Limitations:**

Please refer to "Weaknesses" and "Questions"

**Paper Formatting Concerns:**

No formatting concerns.

**Quality:**

3

**Strengths And Weaknesses:**

Strengths:
1. The topic addressed is meaningful.  As discussed in related work, modifying previous text based 3D editing methods to achieve this functionality causes problematic results. By proposing 3D-OTT, they can achieve faithful 3D texture transferring.
2. The proposed methodology is elegantly simple yet powerful. Specifically, the progressive generation approach successfully resolves the complexities associated with reference views containing original textures. The two gradient guidance strategies presented are clearly articulated and well illustrated by Figure 2.
3. The experimental evaluations are thorough and compelling, featuring extensive numerical and visual comparisons. The ablation study, utilizing two distinct reference images, effectively highlights the robustness and utility of the prompt-tuning gradient guidance.

Weaknesses:
1. The user study currently involves a few participants. Increasing the participant pool will strengthen the reliability and significance of the user evaluation results.
2. This work achieves good editing results, Incorporating a video demonstration would significantly enhance the presentation.

---

> ### Author Rebuttal · Authors · 2025-07-30
>
> We thank the reviewer for the positive feedback and the insightful questions. We address each of your questions in the following sections.
>
> ---
> ### 1. User Study Participant Size
>
> We agree that increasing the number of participants would strengthen the significance of the user study results, and we will expand the participant pool in the revised version. Now, we append results of three new participants in the end.
>
> ---
> ### 2. Request for Video Demonstration
>
> We agree that a video demonstration would enhance the presentation of our results. We will include video demos on our project GitHub and release them alongside the code.
>
> ---
> ### Q1:  Effect of CLIP-Based Token Initialization vs. Random Initialization
>
> We conduct an ablation study to compare two initialization strategies for the prompt used in the CLIP-guided gradient mechanism:
> * Random Initialization: The prompt is randomly initialized and then optimized during fine-tuning.
> * CLIP-Based Initialization: The prompt is initialized by directly projecting the texture difference into CLIP space.
>
> As shown in the table below, initializing the prompt with CLIP-based features leads to significantly better performance in terms of LPIPS (Alex), LPIPS (VGG), and CLIP Score:
>
> |         | LPIPS (Alex) | LPIPS (VGG) | CLIP Score |
> |-----------|:---:|:-----:|:---------:|
> | Random Init.   | 0.1539   | 0.1251    | 0.8574       |
> | Prompt Init.   | 0.1166   | 0.1247   | 0.9333        |
>
> This result indicates that CLIP-based initialization helps the prompt converge faster and guides the diffusion more effectively toward the desired texture characteristics, resulting in better texture fidelity and alignment with the reference.
>
> ---
> ### Q2: Inference Speed Comparison
>
> We compare the inference time of our method against two baselines: the Gaussian Splatting base model (GaussCtrl) and a representative NeRF-based approach. Our method maintains the efficiency benefits of Gaussian Splatting while introducing only modest additional overhead due to our view-consistency and prompt-tuning-based guidance.
>
> In particular, the total additional time required by our proposed guidance components is approximately 8 minutes. Since our full pipeline involves two iterations of image editing, the added cost per iteration is around 4 minutes, which we consider efficient given the improvements in texture fidelity and consistency.
>
> |IN2N|GaussCtrl|Ours|
> |:---: |:---: |:---:|
> |20h:51m:20s|0h:15m:47s|0h:23m:33s|
>
> ---
>
> ### Appended User Study
>
>
>
>
>
>
>
>
>
> We have added 3 new participants as shown in following tables:
> |    User#01       | n2n | gs2gs | gaussctrl | dge | ours |
> |-----------|:---:|:-----:|:---------:|:---:|:----:|
> | pink bear        | 3   | 3     | 1         | 2   | 5    |
> | golden bear      | 2   | 2     | 3         | 2   | 4    |
> | white table      | 1   | 1     | 3         | 2   | 5    |
> | moss table       | 3   | 2     | 1         | 1   | 5    |
> | ice horse        | 3   | 3     | 2         | 1   | 5    |
> | fire horse       | 1   | 3     | 2         | 2   | 5    |
> | lego dinosaur    | 1   | 1     | 2         | 2   | 4    |
> | wooden dinosaur  | 2   | 2     | 2         | 2   | 5    |
> | hawk             | 1   | 2     | 4         | 2   | 5    |
> | hulk             | 2   | 4     | 1         | 2   | 5    |
> | **Avrage**     | 2.1   | 2.3   | 2.1     | 1.8   | 4.8  |
>
> |    User#02       | n2n | gs2gs | gaussctrl | dge | ours |
> |-----------|:---:|:-----:|:---------:|:---:|:----:|
> | pink bear        | 4   | 3     | 1         | 2   | 5    |
> | golden bear      | 2   | 2     | 3         | 2   | 4    |
> | white table      | 1   | 2     | 3         | 2   | 3    |
> | moss table       | 2   | 2     | 1         | 1   | 5    |
> | ice horse        | 2   | 3     | 2         | 1   | 5    |
> | fire horse       | 1   | 3     | 2         | 2   | 5    |
> | lego dinosaur    | 1   | 1     | 2         | 2   | 4    |
> | wooden dinosaur  | 2   | 2     | 2         | 2   | 4    |
> | hawk             | 1   | 2     | 3         | 2   | 5    |
> | hulk             | 2   | 3     | 1         | 2   | 4    |
> | **Average**      | 1.6 | 2.0   | 2.0       | 1.8 | 4.4  |
>
> |    User#03       | n2n | gs2gs | gaussctrl | dge | ours |
> |-----------|:---:|:-----:|:---------:|:---:|:----:|
> | pink bear        | 3   | 3     | 1         | 2   | 4    |
> | golden bear      | 2   | 2     | 3         | 2   | 4    |
> | white table      | 1   | 2     | 3         | 2   | 4    |
> | moss table       | 3   | 2     | 1         | 1   | 5    |
> | ice horse        | 3   | 3     | 2         | 1   | 5    |
> | fire horse       | 2   | 3     | 2         | 2   | 5    |
> | lego dinosaur    | 2   | 2     | 2         | 2   | 5    |
> | wooden dinosaur  | 2   | 2     | 2         | 2   | 5    |
> | hawk             | 3   | 2     | 3         | 2   | 5    |
> | hulk             | 2   | 3     | 2         | 2   | 5    |
> | **Average**      | 2.3 | 2.4   | 2.1       | 1.8 | 4.7    |

---

> > ### Comment · Reviewer_Lzfr · 2025-08-03
> >
> > After reading the authors’ rebuttal, most of my concerns have been addressed. In addition, I believe the ablation studies on Progressive Generation and Flipped Reference Image effectively respond to the concerns raised by Reviewer ZPz8 and Reviewer Eep3. Taking all aspects into account, I find this work sufficiently solid. Based on the authors’ response and the comments from other reviewers, I have decided to raise my score and recommend acceptance.

---

> > > ### Author Response · Authors · 2025-08-04
> > >
> > > Thank you for your encouraging feedback. We appreciate the time you’ve taken throughout the review process. We’re glad you found that the additional ablation studies helped address concerns raised by other reviewers. Thanks again for your support and for recommending acceptance.

---

### Official Review · Reviewer_Eep3 · 2025-06-19

**Clarity:** 1
**Significance:** 3
**Originality:** 3
**Rating:** 4
**Confidence:** 3

**Summary:**

This paper, "3D-OTT: Texture Transfer for 3D Objects from a Single Reference Image," proposes a novel framework for transferring texture from a single 2D reference image to a 3D object represented by 3D Gaussian Splatting. The core aim is to address critical challenges in 3D texture editing, specifically view consistency and the preservation of texture characteristics. The framework comprises three main components: a progressive generation process, view-consistency gradient guidance, and prompt-tuning-based gradient guidance. The first two components are designed to enforce view consistency, while the third ensures the preservation of texture characteristics. The paper demonstrates that 3D-OTT effectively edits the texture, material appearance, and color of 3D objects, outperforming existing text-driven 3D editing methods in both qualitative and quantitative evaluations.

**Questions:**

1. As in Weakness1, the paper states that reference images can be obtained by "directly pasting the reference image onto the 3D object". However, the specific implementation details of this "pasting" or "cropping" process are not clearly described. If this process requires manual selection of the pasting viewpoint or extensive manual intervention.
2. As in weakness2, it lacks an ablation study demonstrating the effect of not using this flip operation in cases where it might be detrimental, which would better clarify its impact.

**Ethical Concerns:**

["NO or VERY MINOR ethics concerns only"]

**Final Justification:**

Thanks the author's reply. I will raise my score to 4.

**Limitations:**

Same as Weakness and Questions.

**Paper Formatting Concerns:**

No formatting concerns.

**Quality:**

3

**Strengths And Weaknesses:**

Strengths:
1. The paper presents compelling qualitative results that visually demonstrate the effectiveness of the proposed 3D-OTT method in faithfully editing 3D objects' textures, materials, and colors based on reference images.
2. The 3D-OTT framework is well-structured with three distinct and clearly defined components: progressive generation, view-consistency gradient guidance, and prompt-tuning-based gradient guidance. These modules are logically designed and intuitively contribute to solving the identified problems of view consistency and texture preservation.

Weakness:
1. Ambiguity in "Depth-based Generation" and "Initial Reference Image" Connection: Section 3.1 describes two ways to obtain reference images: direct pasting of the initial 2D reference image and generating candidate views using a depth-conditioned model. However, the paper lacks clear technical detail on how the depth-conditioned generation process explicitly links to and maintains the specific texture style defined by the initial single 2D reference image. It is unclear how the depth-conditioned model inherently aligns its generated textures with the precise style defined by the initial 2D reference image. If the depth-conditioned model primarily relies on prompts to generate reference images, the underlying process would essentially be text-to-image-to-texture. This contradicts the paper's core premise of "Texture Transfer for 3D Objects from a Single Reference Image," introducing an ambiguity in the main objective and its implementation.
2. The method introduces a horizontally flipped variant of the reference image ($\mathbb{F}(\mathbb{I})_{\tau}$) into the sparse reference set for progressive generation (Section 3.1) to counter error accumulation. This approach inherently assumes some level of symmetry. For 3D objects that are not left-right symmetric, simply flipping the 2D reference image would introduce geometric and textural inconsistencies, potentially leading to erroneous texture transfer and undermining view consistency for asymmetric features. The paper does not discuss this limitation or how it might be addressed.

Minor weakness：
1. In Equation (1) (Section 3.1), the subscript '$e$' in $Attn_{e,e}$ is used without explicit definition within the text, which can lead to minor confusion for readers.
2. In Section 3.2, lines 144-145, the explanation of Equation (2) refers to "differences between guidance with and without the negative prompt."  This phrasing is imprecise and can be confusing, as "negative prompt" typically denotes an explicit textual instruction to avoid certain features in Classifier-Free Guidance, rather than the absence of a conditioning signal (i.e., prompt - none prompt).

---

> ### Author Rebuttal · Authors · 2025-07-30
>
> We appreciate the detailed review and questions. Our responses are provided in the following sections.
>
> ---
> ### 1. Clarification on Connection Between Depth-based Generation and Initial Reference Image
>
> We thank the reviewer for pointing out the ambiguity. We would like to clarify that during progressive generation stage, the depth-based generation process is not primarily driven by prompts, but instead is explicitly conditioned on the visual content of the single 2D reference image and depth. Specifically, we employ a depth-based ControlNet diffusion model that takes both the depth map and the latent embedding of the reference image as inputs. To preserve the texture information from the reference image, we use a k-step partial denoising strategy: the reference image is encoded and reversed into noise for k steps (we set k = 400), and then denoised starting from that noisy state (see lines 46, 128, and 141). This ensures that the exact texture and shape information from the reference image is retained, rather than synthesized anew from a text prompt.
>
> Regarding the reviewer’s concern about whether the process reduces to “text-to-image-to-texture”, we clarify that this is only one optional reference acquisition mode and not central to our method. We further clarify Section 3.1:
>
> * In the manual image-to-image mode, users paste the 2D reference texture image directly onto the object’s silhouette to create a reference image. (Image-to-image-to-texture).
> * Further, we provide an automatic image-to-image mode, where users provide a 2D reference image (which may differ in shape), which is then processed via partial denoising conditioned on depth, without manual intervention. (Image-to-image-to-tetxture).
>
> These reference images, regardless of how they are obtained, are then used in the diffusion process to propagate the texture across views, not regenerated from scratch using a prompt. Text-prompted initial reference generation is only utilized when there is no desired texture.
>
>
> ---
> ### 2. Use of Flipped Reference Image and Asymmetry Concern
>
> Indeed, our method includes a flipped variant $F(I_\tau)$ to improve consistency across distant views (Section 3.1). This design implicitly assumes some degree of left-right symmetry, which does not always hold. We now clarify this limitation and explain how our method mitigates the potential issues:
> * Mitigation via cross-attention: When the object is not symmetric, the flipped image contributes little due to low similarity ($Query \times Key$) in the cross-attention mechanism. The attention weights naturally diminish for dissimilar content, preventing flipped references from introducing significant errors.
> * Ablation results: As shown in our experiment table below on the “bear” scene, the flipped reference improves performance for the symmetric (back) side, while slightly degrading it for the asymmetric (front/head) side. Importantly, the overall performance still improves, as shown by the net gains in LPIPS and stable CLIP scores. Note, lower LPIPS = better and higher CLIP score = better.
> * Optional inclusion: The flipped reference is not strictly required. In scenarios involving strongly asymmetric objects, users or future extensions of our system could disable this component without affecting the core pipeline.
>
> We will include a discussion of this limitation and tradeoff in the revised paper to improve clarity.
>
>
> |           | LPIPS(Alex) | LPIPS(VGG) | CLIP score |
> |-----------|:---:|:-----:|:---------:|
> | Frontside with flipped Ref.      |  0.1488  |   0.1218   |   0.8263      |
> | Frontside W/O flipped Ref.      | 0.1442  |   0.1181   |    0.8263      |
> | Backside with flipped Ref.     |  0.2287  |  0.1867    |   0.8226     |
> | Backside W/O flipped Ref.     |  0.2360  |   0.1973   |     0.8225    |
> | Frontside performance change     |  +0.0046  |  +0.0038    |   0.0     |
> | Backside performance change    |  -0.0073  |   -0.0106   |     +0.0001    |
>
> ---
> ### 3. No Explicit Defination of $e$ in $Attn_{e,e}$
>
>
> Thank you for pointing this out. Index $e$ refers to the image that is currently being edited.
>
> ---
> ### 4. Clarification on “Negative Prompt” Terminology in Eq.(2)
>
> We agree that the terminology in lines 144–145 may be misleading. Our intention was to refer to the unconditioned branch of Classifier-Free Guidance, i.e., the absence of a prompt or a null prompt, not an explicit negative textual instruction. We will revise this phrasing in the final version to use the more precise term “unconditioned prompt” or “null conditioning”, in line with standard CFG terminology.

---

### Official Review · Reviewer_ZPz8 · 2025-06-27

**Clarity:** 3
**Significance:** 2
**Originality:** 3
**Rating:** 5
**Confidence:** 4

**Summary:**

This paper proposes a novel approach for the 2D-to-3D texture transfer task, incorporating a progressive texture generation strategy, multi-view consistent gradient guidance, and fine-tuned prompt-tuning gradient guidance.The progressive texture generation ensures that the guidance image for the current view is derived from adjacent views, enhancing view consistency. The proposed View-Consistency Gradient Guidance steers the generation process toward multi-view consistency while maintaining high-fidelity texture transfer. The method achieves coherent multi-view outputs with realistic texture details.

**Questions:**

1. The title mentions "3D objects" without specifying the modality (e.g., mesh, point cloud, NeRF). Could the authors provide results on other 3D representations?
2. How does the proposed progressive generation fundamentally differ from existing multi-view diffusion methods
3. Can the authors provide ablation experiments for Progressive Generation?

**Ethical Concerns:**

["NO or VERY MINOR ethics concerns only"]

**Final Justification:**

After carefully reviewing the authors' response and the comments from the other reviewers, I believe that most of the initial concerns have been effectively addressed. I have therefore adjusted my rating and recommend that this paper be accepted.

**Limitations:**

yes

**Quality:**

3

**Strengths And Weaknesses:**

Strengths：
 + This paper is well-prepared and easy to follow.
  + The experimental results are impressive.
  + The proposed progressive generation strategy is interesting, and the two guidances are also proven to be effective.

Weakness：
1. The proposed progressive generation resembles existing multi-view diffusion approaches, particularly in its attention mechanism, with insufficient differentiation.
2. The impact of progressive generation remains unverified due to the absence of ablation experiments.
3. In the method section, the method's pipeline lacks clarity, e.g., the transferring process.
4. The symbol *e* in Equation 1 lacks defination, which may confuse readers about its role in the formulation.
5. In Fig.2: （1）The leftmost and bottom panels fail to convey key methodological insights.
（2）The rightmost subfigure lacks detailed explanation in the main text.

---

> ### Author Rebuttal · Authors · 2025-07-30
>
> We thank the reviewer for the constructive feedback and thoughtful questions. Each point is addressed in detail in the following sections.
>
> ---
> ### 1. Clarification on Similarity with Multi-View Diffusion Attention Mechanisms
>
> While our modified generative model may appear similar to existing multi-view diffusion methods at a high level, we emphasize that our task settings, constraints, and diffusion components are fundamentally different.
>
> Specifically, our method takes as input a single 2D reference image and a fixed 3D Gaussian Splatting (3DGS) representation, and performs high-fidelity texture transfer onto this existing geometry. In contrast, multi-view diffusion approaches are typically designed to generate novel views or reconstruct 3D scenes from a few same texture input images, which are not anchored to an explicit 3D representation. These methods synthesize geometry implicitly through learned priors, which makes them unsuitable for editing tasks that require consistency with a given 3D structure. In addition, multi-view diffusion methods assume cross-view texture consistency, which does not hold in the texture transfer setting where the reference image and the 3D representation exhibit different textures. Directly applying them as input leads to collapsed reconstructions.
>
> Regarding the designed diffusion components, our proposed view-consistency guidance and prompt-tuning-based guidance are training-free plugins while finetuning is required for modifications of existing multi-view diffusions.
>
>
>
> We will revise the final version of the paper to make this distinction clearer, particularly in relation to the design and function of our attention mechanism within the 3D editing context.
>
> ---
> ### 2. Justification of Progressive Generation via Ablation Study
>
>
> To evaluate the effectiveness of the proposed progressive generation process, we conduct an ablation study where we disable progressive view propagation and perform editing using only the initial reference image. In this ablated setting, each novel view is generated independently, without leveraging intermediate edited views. The results, averaged across several scenes, are summarized below:
>
> | Averaged | LPIPS (Alex) | LPIPS (VGG) | CLIP Score |
> |---|---:|---:|---:|
> | W/O Prog. Gen. | 0.129 | 0.133 | 0.916 |
> | With Prog. Gen. | 0.116 | 0.125 | 0.933 |
>
> The detailed ablation study results on progressive generation process are listed in following three tables:
>
> |   LPIPS (Alex)       | **Pink bear** | **Golden bear** | **White table** | **Moss table** | **Ice horse** |
> |-----------|:---:|:-----:|:---------:|:---:|:----:|
> | W/O prog. Gen.        | 0.035    | 0.133    | 0.096        | 0.103   | 0.074    |
> | With Prog. Gen.     | 0.038  | 0.111     | 0.047         | 0.094   | 0.056    |
> |          | **Fire horse** | **Lego dinosaur** | **Wooden dinosaur** | **Hawk** |**Hulk** |
> | W/O prog. Gen.        | 0.089   | 0.125     | 0.110       | 0.254   | 0.276    |
> | With Prog. Gen.     | 0.081   | 0.123    | 0.095         | 0.237   | 0.284    |
>
> |   **LPIPS (VGG)**      | **Pink bear** | **Golden bear** | **White table** | **Moss table** | **Ice horse** |
> |-----------|:---:|:-----:|:---------:|:---:|:----:|
> | W/O prog. Gen.        | 0.061  | 0.134    | 0.104         |0.097   | 0.063    |
> | With Prog. Gen.     | 0.073   | 0.116     | 0.068         | 0.096   | 0.056    |
> |          | **Fire horse** | **Lego dinosaur** | **Wooden dinosaur** | **Hawk** |**Hulk** |
> | W/O prog. Gen.        | 0.085   | 0.123     | 0.096        | 0.278   | 0.292    |
> | With Prog. Gen.     | 0.085   | 0.122     | 0.092         | 0.250   | 0.289    |
>
> |   **CLIP Score**      | **Pink bear** | **Golden bear** | **White table** | **Moss table** | **Ice horse** |
> |-----------|:---:|:-----:|:---------:|:---:|:----:|
> | W/O prog. Gen.        | 0.962   | 0.923    | 0.936        | 0.937   | 0.919    |
> | With Prog. Gen.     | 0.926   | 0.939     | 0.968         | 0.963   | 0.922    |
> |          | **Fire horse** | **Lego dinosaur** | **Wooden dinosaur** | **Hawk** |**Hulk** |
> | W/O prog. Gen.        | 0.896   | 0.957     | 0.938        | 0.766   | 0.921    |
> | With Prog. Gen.     | 0.957   | 0.949     | 0.9593         | 0.837   | 0.913    |
>
> These results demonstrate that the progressive generation process significantly improves texture fidelity (lower LPIPS) and semantic alignment with the reference (higher CLIP score). The improvement highlights the importance of propagating texture across neighboring views to enforce view consistency and mitigate artifacts from single-view editing.
>
> ---
> ### 3. Clarification on Pipeline and Texture Transfer Process
>
> We acknowledge that the current presentation of our pipeline in the method section and figure could be clearer. We will revise both the pipeline figure and accompanying text in the revised paper to better illustrate the flow of the texture transfer process.
>
> Specifically, our pipeline proceeds as follows:
> (1) We first obtain the reference image(s), either manually (e.g., by cropping/pasting) or automatically (e.g., via depth-conditioned generation).
> (2) The selected reference image is used to initialize the progressive generation process, where edited images for multiple views are produced using our proposed view-consistency and prompt-tuning-based gradient guidance.
> (3) These high-quality edited views are then used to fine-tune the 3D Gaussian Splatting (3DGS) representation.
> (4) We re-render the updated 3D object into new images and repeat steps (2) and (3) for further refinement.
>
> Typically, two iterations are sufficient to achieve high-fidelity, view-consistent texture transfer across the full object. We will incorporate this clarified explanation and an updated pipeline figure in the revised paper to improve the overall readability of the method section.
>
> ---
> ### 4. No Definition of $e$
> Thanks for pointing this out. Index $e$ refers to the image that is currently being edited.
>
> ---
> ### 5. Clarification of Figure 2 (Pipeline)
>
>  We agree that certain components of Figure 2 could be made more informative. Specifically:
>
> (1) We will revise the leftmost and bottom panels to better convey the reference image acquisition process and the rationale behind using flipped variants and sparse cross-attention.
>
> (2) We also acknowledge that the rightmost subfigure (showing the iterative editing loop) lacks sufficient explanation in the main text. We will revise the manuscript to clarify that:
> * The reference image is obtained either manually or via depth-conditioned generation.
> * It is used to initiate the progressive generation stage, guided by both view-consistency and prompt-tuning-based gradient guidance.
> * The resulting edited images are then used to fine-tune the 3D Gaussian Splatting representation.
> * This process is iterative: we re-render the updated 3D objects and apply the same editing process again. Typically, two iterations are sufficient to achieve high-quality texture transfer.
>
> We will revise both the figure and the method description accordingly to ensure that these steps are clearly communicated.
>
> ---
> ### Q1: Clarification on “3D Objects” and Generalizability Across Modalities
>
> We agree that the term “3D objects” in the title may be too broad, and we are open to revising it to “3DGS-OTT: Texture Transfer for 3D Gaussian Splatting from a Single Reference Image” to more accurately reflect our current implementation.
>
> While our experiments focus on 3D Gaussian Splatting (3DGS) due to its compatibility with GaussCtrl and efficient rendering, the core contributions of our method: progressive generation, view-consistency gradient guidance, and prompt-tuning-based gradient guidance, are representation-agnostic. They operate independently of the underlying 3D format and can be adapted to other modalities such as NeRF, point clouds, or meshes, by treating diffusion-based generation and 3D model finetuning as two decoupled stages.
>
> We also plan to release a modular version of our pipeline that includes only the diffusion-based editing stage, allowing users to integrate it with other 3D representations depending on their downstream applications.
>
> ---
> ### Q2 (Multi-view Diffusion Methods) & Q3 (Ablation for Progressive Generation):
>
> We believe these questions are addressed in our responses to previous discussions on the distinction between our progressive generation and existing multi-view diffusion methods, and where we provide an ablation study validating the effectiveness of the progressive generation process.

---

> > ### Comment · Reviewer_ZPz8 · 2025-08-04
> > **Official Comment by Reviewer ZPz8**
> >
> > Thank you to the authors for their detailed and convincing rebuttal. After reviewing their response, I believe my major concerns have been well resolved. For the paper's title, I would recommend "3DGS-OTT: Texture Transfer for 3D Gaussian Splatting from a Single Reference Image." This version is more precise, as the current work has only demonstrated effectiveness on 3D Gaussian Splatting and not on other representations.

---

> ### Author Response · Authors · 2025-08-04
> **Thank You for the Positive Feedback and Consideration**
>
> Thank you for your positive feedback. We’re glad to hear that your major concerns have been addressed, including (i) the difference between Progressive Generation and existing multi-view diffusion methods, and (ii) the ablation study for Progressive Generation.
>
> We also agree with your suggestions regarding the title, Figure 2, and other points, and plan to incorporate them into the updated version. Thanks again for taking the time to review our rebuttal. We appreciate your support during the review process.
>
> Should the revisions meet your expectations, we would greatly appreciate your consideration in raising your final score.

---

### Official Review · Reviewer_qSJ7 · 2025-07-02

**Clarity:** 3
**Significance:** 2
**Originality:** 2
**Rating:** 4
**Confidence:** 4

**Summary:**

The paper presents 3D-OTT, a novel method for transferring textures from a single 2D reference image onto a 3D object represented via Gaussian Splatting. To address key challenges like view inconsistency and poor texture fidelity in unseen views, the method leverages a progressive generation process that starts from the reference view and incrementally propagates edits to neighboring views. To improve consistency it introduces a view-consistency gradient guidance to enforce multi-view coherence during diffusion, and prompt-tuning-based gradient guidance that learns a token capturing the texture difference between the original and reference, preserving style across unseen views.

**Questions:**

1. The method assumes geometric similarity between the reference image and the 3D object. What happens when this assumption fails (e.g., different pose, incorrect scale)?

**Ethical Concerns:**

["NO or VERY MINOR ethics concerns only"]

**Final Justification:**

The authors’ rebuttal has anddressed my comments and provided an explanation to some minor quality concerns. Give that I’m inclined to recommend acceptance for this paper.

**Limitations:**

Yes.

**Paper Formatting Concerns:**

No concerns.

**Quality:**

2

**Strengths And Weaknesses:**

The paper addresses a relevant problem in vision. Quality is reasonable, with extensive comparisons and ablations. The method clearly outperforms prior baselines in both quantitative metrics and visual fidelity. However, it’s hard to ignore that much of the method stands on stitching together known ideas with limited methodological novelty. The paper is clear enough with the method being effectively described. The authors ablate the main methodological components showing their impact on performance. Originality is the weakest dimension, build upon existing techniques in a relatively incremental way. On closer inspection results present some artifacts especially in the form of a dark border around the edited area.

---

> ### Author Rebuttal · Authors · 2025-07-30
>
> We thank the reviewer for the thoughtful comments and the positive feedback. Detailed responses for each question are provided in the following sections.
>
> ---
> ### 1. Originality Concern
>
> While our method builds upon existing diffusion and prompt-based techniques, we believe our contributions offer meaningful and novel extensions for the 3D texture editing setting. In particular, we propose three key innovations:
>
> a) Progressive Generation Process:
> To the best of our knowledge, our method is the first to introduce a progressive generation strategy in the context of 3D texture editing. Unlike existing multi-view diffusion methods that generate all views in parallel, our approach incrementally propagates texture edits across views by using previously edited views as reference inputs. This step-wise refinement promotes spatial coherence, enhances view consistency, and supports integration with 3D Gaussian Splatting.
>
> b) View-Consistency Gradient Guidance:
> We propose a novel classifier-free guidance format formulation that explicitly incorporates multi-view coherence signals. This is achieved by comparing multi-view-conditional and unconditional noise predictions, allowing us to extract and reinforce view-consistent information during the diffusion process. To our knowledge, this idea has not been applied in diffusion-based 3D editing.
>
> c)  Prompt-Tuning-Based Gradient Guidance:
> While prompt tuning is a well-established technique, we propose a novel application: encoding the texture difference between the original object and the reference image as a learned prompt token. We further extract the texture difference information by the proposed equation and apply it to guide the diffusion process in a consistent stylistic direction. We believe this is the first use of prompt tuning for controlling texture appearance in a 3D-aware diffusion editing pipeline.
>
> In summary, we introduce three new mechanisms that are carefully tailored to address the unique challenges of 3D texture transfer, specifically view consistency and faithful texture preservation, which standard components fail to address. We believe these innovations constitute a non-trivial contribution to the field.
>
> ---
>
> ### 2. Dark Border Artifact Around Edited Regions
>
> The dark border artifact is primarily caused by the use of language-based object segmentation methods (e.g., LangSAM) to generate masks for isolating objects in the reference views. These segmentation methods often include a narrow band of background pixels near object boundaries due to imperfect boundary localization.
>
> As a result, during the diffusion-based editing stage, this narrow band is misinterpreted as valid texture content, leading to the appearance of dark borders in the final renderings. To mitigate this, we apply dilation and Gaussian blurring to soften mask boundaries and reduce the visibility of such artifacts. While these steps alleviate the issue to some extent, traces may still remain in some challenging cases.
>
> We acknowledge this segmentation field limitation and plan to address it more thoroughly in future work by incorporating more advanced and boundary-aware segmentation techniques.
>
> ---
> ### Q1: Geometry Differences Between Reference Images and 3D Objects
>
> Geometry mismatches between the 2D reference image and the 3D object can occur when the reference is generated via a depth-conditioned diffusion model with a small condition scale factor. We address this in two common scenarios:
>
> (a) Slightly Different Geometry:
> In this case, the reference image shows small geometric deviations (e.g., slight pose or scale differences), such as a standing bear with legs in a slightly different position. Our method is robust to such differences due to the following mechanisms:
> * During progressive generation, texture features are extracted via cross-attention while being constrained by both the depth map and the partially reversed latent features which preserve appearance clues of the original image. This encourages edits that align with both the 2D reference and 3D structure.
> * During 3D fine-tuning, view-consistent textures are enforced across adjacent edited views. Overlapping regions from neighboring views iteratively correct geometry discrepancies and reinforce accurate texture transfer.
>
> (b) Significantly Different Geometry:
> If the reference image shows a completely different shape (e.g., a running bear instead of a standing one), transfer quality may degrade, as discussed in Section 4.4 of our paper. However, such poor references typically arise from failure in the depth-conditioned option generation stage. In practice, users can easily detect these cases during the reference selection stage (which is interactive), and regenerate better references using our partial denoising strategy, which conditions on depth and corrects shape biases during generation.
>
> In summary, our method is robust to minor geometric mismatches and provides mechanisms to mitigate large discrepancies through reference regeneration and iterative correction during 3D fine-tuning.

---

> ### Author Response · Authors · 2025-08-06
>
> Dear Reviewer qSJ7, thank you for your initial review. They were insightful and helped us strengthen the paper. We just want to check if our response has addressed your points. If any additional clarification or supporting results are needed, we’d be happy to provide them. We appreciate your positive feedback, and we hope the rebuttal we’ve provided has resolved your concerns.

---

> ### Comment · Area_Chair_evhA · 2025-08-09
>
> Thank you for your efforts of reviewing! Please remember to read the rebuttal as soon as possible, and discuss with the authors for questions not resolved. According to the guideline this year, reviewers need to participate in discussions with authors before submitting “Mandatory Acknowledgement”.

---

### Note · Authors · 2025-08-12

We thank all reviewers and the AC for their time, constructive feedback, and engagement. We are pleased that two reviewers (qSJ7, Lzfr) gave supportive scores for our work from the outset. After the discussion, Lzfr expressed that they will raise their rating, and Zpz8 stated that their major concerns have been well resolved. We are committed to incorporating their suggestions into the final version.

We also thank Reviewer qSJ7 for the initial positive score and suggestion. While qSJ7 did not join the discussion phase, we have addressed their concerns.

We respectfully request that the AC and reviewers consider the overall support and the substantial clarifications provided during the rebuttal and discussion when making the final decision.

---

### Decision · Program_Chairs · 2025-09-17

**Decision:**

Accept (poster)

**Comment:**

The paper proposes a method for transferring texture from a reference image to a 3D object using diffusion models. The approach progressively edits multi-view images, leveraging a reference set (previous views, the reference view, and its flipped version), enhanced classifier-free guidance, and a learned token that encodes texture discrepancies to guide denoising toward the desired style. The method demonstrates strong qualitative results.

However, several weaknesses remain:
- Clarity and presentation: The writing contains numerous typos and unclear descriptions (as noted by Reviewers ZPz8 and Eep3). Many sections (for method) are not self-contained, which may hinder accessibility to general readers. Additionally, the lack of a clear teaser figure limits understanding of the pipeline (Fig. 2 is not sufficiently informative).
- Assumptions on reference images: Despite the claim of using a single reference image, the method relies on restrictive assumptions (e.g., requiring consistent geometry with varying textures, and the need for text prompts), as highlighted by Reviewer ZPz8.
- Evaluation concerns: Results without progressive generation (as shown in the rebuttal) already outperform baselines, raising concerns about whether the baselines are too weak. Furthermore, performance discrepancies between “ours” in Tables 1 and 2 are confusing.

The rebuttal provides additional ablations and clarifies the pipeline, which improves the submission. Overall, the paper presents clear technical modules and compelling qualitative results, but clarity, writing quality, and evaluation remain areas for improvement.